# Fabric Defect Detection Based on Illumination Correction and Visual Salient Features

**DOI:** 10.3390/s20185147

**Published:** 2020-09-09

**Authors:** Lan Di, Hanbin Long, Jiuzhen Liang

**Affiliations:** 1School of Artificial Intelligence and Computer, Jiangnan University, Wuxi 214122, China; dilan@jiangnan.edu.cn (L.D.); 6191611032@stu.jiangnan.edu.cn (H.L.); 2School of Information and Engineering, Changzhou University, Changzhou 213164, China

**Keywords:** illumination correction, quaternion representation, 2D-FRFT, visual salient features, fabric defect detection

## Abstract

Aiming at the influence of uneven illumination on fabric feature extraction and the limitations of traditional frequency-based visual saliency algorithms, we propose a fabric defect detection method based on the combination of illumination correction and visual salient features—(1) Construct a multi-scale side window box (MS-BOX) filter to extract the illumination component of the image, then use the constructed two-dimensional gamma correction function to perform illumination correction on the image in the global angle, and finally enhance the local contrast of the image in the local angle; (2) Use the $L_{0}$ gradient minimization method to remove the background texture of fabric images and highlight the defects; (3) Represent the fabric image as a quaternion image, where each pixel in the image is represented by a quaternion consisting of color, intensity and edge characteristics. The two-dimensional fractional Fourier transform (2D-FRFT) is used to obtain the saliency map of the quaternion image. Experiments show that our method has a higher overall recall rate for defect detection of star-patterned, box-patterned, and dot-patterned fabrics, and the overall recall-precision effect is better than other existing methods.

## 1. Introduction

In the process of fabric production, defect detection is very important to the quality control of fabrics. Nowadays, the defect detection of fabrics is mainly aimed at two kinds of fabrics—(1) those with no complex texture pattern, simple structure, mostly solid-color fabrics and (2) those with complex patterns, including pariodic fabrics.

For the first kind of fabric, the research methods have been mature. The main subtypes are—(1) statistical methods, such as the co-occurrence matrix method [1] and the morphological method [2]; (2) spectral methods, such as Fourier transform [3], wavelet transformation [4], Gabor filtering [5]; (3) model methods, such as Markov random field model [6]; (4) deep learning methods, which have been widely used in computer vision [7,8], many researchers have also begun to use deep networks to detect fabric defects, such as neural networks [9], Mobile-Unet [10]; statistical and spectral methods when the area is too large or defects are too small, caused by error. The model method needs to define the model in advance, and different models need to be defined for different types of defects, which is not universal. The deep learning method needs a large number of samples as the training set, and training parameters take a long time. When the defect is close to the texture backgroud, Mobile-Unet [10] cannot detect the details of the defect well. For the second kind of fabric, there are few mature and available methods, mainly including Wavelet Preprocessed Golden Image Subtraction (WGIS) [11], Bollinger Bands (BB) [12], Regular Bands (RB) [13], Elo Ranking (ER) [14], and Similarity Relation (SR) [15]. The calculation process of these methods is complicated, and the precision and recall of detecting defects need to be improved. Low-rank decomposition [16,17] obtains the sparse defect part by decomposing the image into low-rank matrix and sparse matrix. This kind of method needs to construct the feature matrix, and then decompose the feature matrix with low-rank. The detection effect depends on the feature selection, and the robustness is poor and time-consuming.

Visual saliency algorithm simulates human vision, detects the difference between fabric defects and the normal background from a visual perspective, and separates the defect from the background to complete the defect detection. Frequency tuned salient region detection (FT) [18] transforms the defect image from RGB color space to CIE Lab color space, and regards the defect as an area with saliency characteristics by using the difference between the defect and background color and brightness. Low level visual saliency detection algorithm based on wavelet transform [19] uses multi-directional wavelet to carry out two-dimensional discrete wavelet transform on three channels of CIE Lab color space, and uses the way of global and local feature fusion to form the saliency map of defect area. The FT algorithm only considers the global features, and the detection is poor when the defect is close to the background color. The saliency algorithm based on wavelet transform [19] is more complex for the defect detection time of periodic fabrics, the calculation process is more cumbersome, and the adaptability to different types of defects needs to be improved. The spectral residual method based on Fourier transform [20] only considers the saliency of images in the frequency domain and ignores the saliency of images in the time domain and the space domain.

In the process of collecting fabric defect images, it is easy to be affected by uneven illumination, which increases the difficulty of image feature extraction and improves the error detection rate of defect detection. The traditional histogram equalization method and self quotient image method [21] are easy to make the image over enhanced and the effect of illumination correction is not good. In recent years, homomorphic filtering methods [22] and Retinex based methods [23] have been widely used in image illumination correction.

To overcome the above issues, we propose a method of fabric defect detection based on the combination of illumination correction and visual salient features. Firstly, to solve the problem that traditional illumination correction methods tend to overenhance the image, the illumination component of the fabric image is extracted by using the multi-scale side window box filter (MS-BOX) in the global angle, then enhance the local contrast of the fabric image in the local angle. Secondly, to solve the interference of texture background, we use the $L_{0}$ gradient minimization method to remove the background texture of fabric image. Then, the fabric image is represented as a quaternion image. Each pixel in the image is represented by a quaternion composed of color, intensity, and edge features. Finally, to solve the limitation of traditional frequency domain methods only considering the salient characteristics of the defect in the frequency domain, the salieny map of the fabric image is obtained by using the two-dimensional fractional Fourier transform. The contributions of this paper are summarized as follows:Different from traditional methods that only perform illumination correction locally or globally, our method performs illumination correction on the fabric image in both global and local angles.Different from the traditional method of constructing quaternion images, we choose a color space that is more suitable for fabric images, improve the robustness of the intensity feature channel, and replace the motion feature channel with edge feature channel.Different from the traditional frequency domain method using simple Fourier transform to obtain the saliency map, we use the two-dimensional fractional Fourier transform to obtain the saliency map of the quaternion image.

The remainder of this paper is organized as follows—in Section 2, the work related to illumination correction and visual salient feature are briefly described. In Section 3, we propose a fabric defect detection method based on the combination of illumination correction and visual salient features and discuss the implementation details. In Section 4, we evaluate the performance of our method on a standard data and compare it with existing representative methods WGIS, Mobile-Unet, SHF, ER, CDPA, and SR. Finally, Section 5 concludes the paper.

## 2. Related Work

### 2.1. Illumination Correction

According to Retinex theory [23], an image q(x,y) can be divided into two different images: the reflected object image r(x,y) and the illumination image i(x,y)
(1)q(x,y)=r(x,y)·i(x,y).

The multi-scale rolling guidance filter (RGF) [24] is widely used to extract the illumination component of an image. Ying et al. introduced a Exposure Fusion Framework (EFF) [25] and a Bio-Inspired Multi-Exposure Fusion Framework (BIMEFF) [26] for low-light image enhancement, where the enhanced result is obtained by fusing the input image and the synthetic image according to the weight matrix. Ren et al. [27] proposed a Joint low-light Enhancement and Denoising (JED) strategy, where enforced the spatial smoothness on each component and skilfully made use of weight matrices to suppress the noise and improve the contrast. In order to solve the problem of low visibility, Guo et al. [28] proposed a simple yet effective low-light image enhancement (LIME) method. Lore et al. [29] proposed a deep autoencoder approach to low-light image enhancement (LLNet). But the illumination correction effect of these methods for fabric images needs to be improved.

### 2.2. Visual Salient Feature

Visual saliency is a fundamental problem in image processing, pattern recognition, and computer vision. In recent years, many scholars have used visual salient features to detect fabric defects.

Li et al. [30] introduced a Saliency Histogram Features (SHF) method, in which they extrcted and selectd saliency histogram features to discriminate between the defective and defect-free fabric images. Zhang et al. [31] proposed a Color Dissimilarity and Positional Aggregation (CDPA) method, in which they measured the defect value based on the color difference and the position distance between similar color blocks. This kind of method has achieved certain effect, but its real-time performance is not good. The general residual analysis using Fourier transform is relatively simple and fast. On this basis, Guo et al. [32] used the phase spectrum instead of the original amplitude spectrum.

## 3. Methods

The steps of our method include illumination correction, texture background removal, saliency map generation and segmentation. Figure 1 is the framework of the proposed our method.

### 3.1. Illumination Correction

#### 3.1.1. Illumination Correction in the Global Angle

The traditional method of extracting image illumination component by multi-scale rolling guidance filter [24] has some defects, for example, damaged image edge and illumination components appear halo phenomenon. The side window filter (SWF) [33] can preserve the edge of the image very well, so we use multi-scale side window box filter to extract the illumination components of fabric images. Considering that HSV color space is more consistent with the visual characteristics of human eyes, and the hue (H), saturation (S), and value (V) in HSV color space are independent of each other, the operation of V will not affect the color information of the image. So we choose to convert the image from RGB color space to HSV color space.

The definition of a side window is shown in Figure 2a. θ is the angle between horizontal line and the window, *r* represents the radius of the window, $\rho \in \left\{0,r\right\}$, and (x,y) is the position of the pixel *i*. By fixing (x,y) and changing θ, we are able to adjust the direction of the window and align its side with *i*.

In order to simplify the process, we adopt the proposal of the paper [33], and define eight side windows only in the discrete cases, as shown in Figure 2b–d. These eight windows defined here correspond to $\theta =k\times \frac{\pi }{2}$, k∈[0,3]. By setting ρ=r, we can get the down(D), right(R), up(U) and left(L) side windows, named as $\omega_{i}^{D}$, $\omega_{i}^{R}$, $\omega_{i}^{U}$ and $\omega_{i}^{L}$. They are aligned *i* with their sides. By setting ρ=0, we can get the southwest(SW), southeast(SE), northeast(NE) and northwest(NW) side windows, named as $\omega_{i}^{SW}$, $\omega_{i}^{SE}$, $\omega_{i}^{NE}$ and $\omega_{i}^{NW}$. They are aligned *i* with their corners.

By applying the filtering kernel *F* to each side window, we can get eight different outputs, named as $I_{i}^{{}^{\prime }\theta ,\rho }$, where $\theta =k\times \frac{\pi }{2}$, k∈[0,3] and $\rho \in \left\{0,r\right\}$
(2)$$I_{i}^{{}^{\prime }\theta ,\rho }=F\left(q_{i},\theta ,\rho ,r\right),$$
where $q_{i}$ and $I_{i}$ are the intensities of the input image *q* and the output image *I* at location *i*, respectively. In order to preserve the edges, we want to minimize the distance between the input and the output at the edge. Consequently, we select the side window output with the minimum $L_{2}$ distance to the input intensity as the final output,
(3)$$I_{SWF}^{{}^{\prime }}=\underset{\forall I_{i}^{{}^{\prime }\theta ,\rho ,r}}{\arg\min}{∥q_{i}-I_{i}^{{}^{\prime }\theta ,\rho ,r}∥}_{2}^{2},$$
where $I_{SWF}^{{}^{\prime }}$ is the output of SWF.

In order to enhance the robustness of the original SWF, we build a multi-scale SWF by changing the window radius *r*, and introduce the box filter (BOX) into the multi-scale SWF. That is to say, *F* in Equation (Equation 2) is averaging and the resulting filter is called multi-scale side window box filter (MS-BOX),
(4)$$I_{MS-BOX}^{{}^{\prime }}=\sum_{j=1}^{n}\frac{1}{n}\left[\underset{\forall I_{i}^{{}^{\prime }\theta ,\rho ,r_{j}}}{\arg\min}{∥q_{i}-I_{i}^{{}^{\prime }\theta ,\rho ,r_{j}}∥}_{2}^{2}\right],$$
where $I_{MS-BOX}^{{}^{\prime }}$ is the outout of MS-BOX. *n* is the number of scales. In this paper, n=3.

By convoluting the value component V(x,y) of the image with MS−BOX(x,y), the estimated value of the illumination component I(x,y) can be obtained. The results are as follows:(5)I(x,y)=MS−BOX(x,y)·V(x,y).

After extracting the illumination component of the image, the gamma correction function can be constructed according to the distribution characteristics of the illumination component. The expression of the two-dimensional gamma correction function constructed in this paper is as follows:(6)$$O\left(x,y\right)=255{\left(\frac{V(x,y)}{255}\right)}^{\lambda },\lambda ={\left(\frac{1}{2}\right)}^{\frac{m-I(x,y)}{m}},$$
where O(x,y) is the brightness value of the output image after correction, λ is the index value used for brightness enhancement, which contains the characteristics of the illumination component of the image, *m* is the mean value of the estimated value of the illumination component I(x,y).

#### 3.1.2. Enhance the Contrast in the Local Angle

Local Contrast Enhancement (LCE) algorithm [34] can effectively improve the visualization of detail features and keep the original details of the image as much as possible. The transformation equation of LCE algorithm is as follows:(7)$$Y\left(m,n\right)=\left\{\begin{array}{l} \log\left(\frac{L(m,n)}{\bar{L(m,n)}}\right),L\left(m,n\right)>\theta ,\bar{L(m,n)}>\theta \\ 0,otherwise \end{array}\right.$$
(8)$$\bar{L(m,n)}=\frac{1}{N}\sum_{i,j\in \Omega }^{}L(m+i,n+j),$$
where Θ is a predefined threshold and L(m,n) is the gray value at the pixel (m,n). $\bar{L(m,n)}$ represents the local gray value of the pixel (m,n) in the domain. Y(m,n) represents the adjustment gray value of the pixel (m,n). In this experiment, we use the domain of 5 × 5, where *N* is the total number of pixels in the selected domain. Since the local value of Equation (Equation 7) can be positive or negative, it is necessary to normalize it:(9)$$f\left(m,n\right)=\frac{Y\left(m,n\right)-Y_{\min}}{Y_{\max}-Y_{\min}}\times 255.$$

We combine the above two methods to achieve illumination correction of fabric images in both global and local angles.

### 3.2. Extract Visual Salient Features of Image

#### 3.2.1. Background Texture Smooth by the $L_{0}$ Gradient Minimization(LGM)

Because of the diversity of pattern and texture, it usually brings great difficulties to fabric defect detection. In recent years, due to the fast and effective of the LGM algorithm [35], LGM has been used by many scholars to remove texture. The LGM can not only smooth the background texture, but also retain the key information of the image. In brief, the LGM preserves the important edge parts of the image by adding the steepness of the transition part of the image while removing low-amplitude parts. Let *I* be the input image, the result of the LGM is *S*. The partial derivatives of the smoothed image at *p* in the *x* and *y* directions are defined as $\partial_{x}S_{p}$ and $\partial_{y}S_{p}$ respectively. Therefore, the gradient of smoothed output *S* at pixel *p* can be expressed as:(10)$$\nabla S_{p}={\left(\partial_{x}S_{p},\partial_{y}S_{p}\right)}^{T}.$$

So the image $L_{0}$ gradient specific objective function can be expressed as:(11)$$\min\left\{\sum_{p}{\left|S_{p}-I_{p}\right|}^{2}+\beta {\left|\nabla S_{p}-h\right|}^{2}+\lambda {\left|h\right|}_{0}\right\},$$
where λ is non-negative parameter, which affects the degree of image smoothing. *h* is auxiliary variable and β is an adaptive parameter. By alternatively computing *h* and *S*, we can get the smoothed output result.

As shown in Figure 3a, the input image has a complex texture structure, and its mesh diagram is shown in Figure 4a. After the LGM algorithm, the complex texture information of input image is smoothed, and the output result is shown in Figure 3b. It is noted that the important edges of the defect are preserved, and the defect is more visible, and the mesh diagram is shown in Figure 4b.

#### 3.2.2. Creation of a Quaternion Image

Saliency detection method based on quaternion [32] represents each pixel with a quaternion consisting of color, intensity, and motion features. Compared with RGB color space, CIE Luv color space is more suitable for fabric defect detection with single color. L represents brightness, and u and v represent chroma. Therefore, we change the input image *I* from RGB color space to CIE Luv color space. Let *l*, *u* and *v* represent different channels of image *I* in CIE Luv color space. Equations (12)–(15) create four broadly-tuned color channels:(12)L=l−(u+v)/2
(13)U=u−(l+v)/2
(14)V=v−(l+u)/2
(15)$$Y=\left(l+u\right)/2-\left|l-u\right|/2-v.$$

In human brain, there exists a ‘color opponent-component’ system [36]. In the center of receptive fields, neurons are excited by one color or chroma. The opposite chroma channels are obtained by Equations (16) and (17).
(16)LU=L−U
(17)VY=V−Y.

In order to further reduce the non-saliency of the color and strengthen its biological rationality, we adjusted the intensity channel *F*,
(18)$$F=(\bar{l}+\bar{u}+\bar{v})/3,$$
where $\bar{l}=l-l_{m}$, $\bar{u}=u-u_{m}$, $\bar{v}=v-v_{m}$. $l_{m}$, $u_{m}$, and $v_{m}$ are the mean value of *l*, *u*, and *v* respectively.

Since we are dealing with static images without motion features, we use Canny operator to extract edge features *E* instead of motion features, According to the above four feature channels, the quaternion image *q* is defined as follows:(19)$$q=f_{1}+f_{2}\cdot \mu_{2}$$
(20)$$f_{1}=E+LU\cdot \mu_{1}$$
(21)$$f_{2}=VY+F\cdot \mu_{1},$$
where $\mu_{i}$, i=1,2 satisfies $\mu_{i}^{2}=-1$, $\mu_{1}⊥\mu_{2}$.

#### 3.2.3. Using 2-D Fractional Fourier Transform to Obtain Saliency Map

The fractional Fourier transform (FRFT) is a generalized form of the traditional Fourier transform. The result of the transform contains the information of signal time and frequency domains. For an input signal x(t), the FRFT is as follows:(22)$$X_{\alpha }\left(u\right)=\left\{F^{\alpha }\left[x(t)\right]\right\}\left(u\right)=\int_{-\infty }^{\infty }x\left(t\right)K_{\alpha }\left(t,u\right)dt$$
(23)$$K_{\alpha }=\sqrt{\frac{1-j\cot\alpha }{2\pi }\exp\left(j\frac{t^{2}+u^{2}}{2}\cot\alpha -tu\csc\alpha \right)},$$
where α is the rotation angle when the signal rotates to the frequency axis, α=p·π/2, and *p* is the transformation order of fractional Fourier transform. It can be seen from Equations (22) and (23) that when p=1, the rotation angle is π/2, and the fractional Fourier transform degenerates into the traditional Fourier transform; when p=4n, the rotation angle is an integral multiple of 0 or 2, and the result of the fractional Fourier transform is the signal itself; when *p* is a fraction, the rotation angle is between 0 and π/2, and the signal is rotated between the time axis and the frequency axis, In this case, the results of FRFT can describe the signal characteristics from both the time and the frequency domain. Figure 5 shows the transform domain of fractional Fourier transform, where axis *t* represents the time axis and axis ε represents the frequency axis.

For a two-dimensional signal x(s,t), its two-dimensional fractional Fourier transform (2D-FRFT) is defined as:(24)$$X_{\alpha ,\beta }\left(u,v\right)={F_{\alpha }}^{t\to v}\left\{{F_{\beta }}^{s\to u}\left[x(s,t)\right]\right\},$$
where α and β represent two independent fractional rotation angles in two-dimensional space, and the two-dimensional transformation result of signal x(s,t) is equal to two successive fractional Fourier transforms of the signal with parameters α and β respectively. In this work, we set both α and β to 0.9.

The transform kernel of two-dimensional fractional Fourier transform can be defined as follows:(25)$$K_{\left(\alpha ,\beta \right)}=K_{\alpha }\times K_{\beta },$$
where α and β are discrete forms of kernel functions of fractional Fourier transform.

For a discrete two-dimensional signal f(m,n), the discrete two-dimensional fractional Fourier transform at point (m,n) is as follows:(26)$$X_{\left(\alpha ,\beta \right)}\left(m,n\right)=\sum_{p=0}^{M-1}\sum_{q=0}^{N-1}x\left(p,q\right)K_{\left(\alpha ,\beta \right)}\left(p,q,m,n\right).$$

The inverse discrete two-dimensional fractional Fourier transform at point (m,n) is as follows:(27)$$x\left(p,q\right)=\sum_{m=0}^{M-1}\sum_{n=0}^{N-1}X_{\left(\alpha ,\beta \right)}\left(m,n\right)K_{\left(-\alpha ,-\beta \right)}\left(p,q,m,n\right).$$

#### 3.2.4. Generation of Saliency Map

The 2D-FRFT of Equation (Equation 19) can be written as:(28)$$Q\left(u,v\right)=X_{1}\left(u,v\right)+X_{2}\left(u,v\right)\mu_{2},$$
where $X_{i}\left(u,v\right),i=1,2$ is the two-dimensional fractional Fourier transform of $f_{i}$.

Q(u,v) can be represented in polar form as:(29)$$Q\left(u,v\right)=∥Q(u,v)∥e^{\mu \phi },$$
where $∥•∥$ is the amplitude spectrum, ϕ is the phase spectrum and μ is a unit pure quaternion.

Calculate the inverse two-dimensional fractional Fourier transform of Q(u,v) using Equation (Equation 27), the result is written as $\bar{Q}\left(u,v\right)$.

The final saliency map is obtained by Equation (Equation 30),
(30)$$S=g*{∥\bar{Q}\left(u,v\right)∥}^{2},$$
where *g* is a 2D gaussian filter (σ=2.5)

We use the region growing method to segment the saliency map and separate the defect from the background. Finally, the morphological treatment of the saliency map is carried out to remove the noise points that are easily caused by misdetection.

#### 3.2.5. Computation Cost Analysis

The computational cost of our method is mainly affected by the following work: illumination correction, use LGM to remove texture background, construct a quaternion image and saliency map generation.

Let N=M×N, *M* and *N* are the width and height of the input image respectively, the illumination correction process is a linear calculation process, so its computational complexity is O(N); the computational complexity of LGM is mainly determined by Equation (Equation 11), so its computational complexity is O(NlogN); the process of constructing a quaternion image is also a linear calculation process, so its computational complexity is O(N); the computational complexity of saliency map generation is mainly determined by Equation (Equation 28), so its computational complexity is O(NlogN). Therefore, the computation complexity of our method is as follows:(31)$$\begin{array}{cc} T(method) & =O(N)+O(N\log N)+O(N)+O(N\log N)\\ & =O(N\log N) \end{array}.$$

Besides, the space complexity of our method is O(C×N), where *C* is a constant.

## 4. Experiments and Performance Evaluation

In this section, our work is performed by using in total of 50 images provided by the automation laboratory fabric database of Hong Kong University. More specifically, 15 defect images of size 256 × 256 are from the box-patterned fabric database, 15 defect images from the star-patterned fabric database and 20 defect images from the dot-patterned fabric database. In addition, all defect images have corresponding binary ground truth images, with a value of 1 for defective objects and 0 for defect-free objects. WGIS [11] (2005), Mobile-Unet [10] (2020), SHF [30] (2019), ER [14] (2016), CDPA [31] (2018) and SR [15] (2017) are implemented for comparison. The experiments are performed on a personal computer with an Intel Core i5-8300H processor and 8 GB memory. The testing codes are implemented in Matlab 2019a.

### 4.1. Analysis of Experimental Results of Different Illumination Correction Methods

Different illumination correction methods have different correction effects on fabric images, and comparison experiments of different methods are carried out for this problem. The scale factors *r* of the multi-scale MS-BOX are 3, 5, and 7, respectively. Figure 6 shows the illumination component extraction effect comparison of multi-scale RGF [24] (2014) and multi-scale MS-BOX. Figure 7 shows the illumination correction effect comparison of BIMEFF [26] (2017), JED [27] (2018), LIME [28] (2017), EFF [25] (2017), LLNet [29] (2017) and Ours.

As the Figure 6 shows, compared with the multi-scale RGF, the multi-scale MS-BOX can eliminate the halo phenomenon in the illumination component image to a certain extent. This is because in the multi-scale MS-BOX, the edge information of the image is preserved in the filtering process. The illumination component extracted by the multi-scale MS-BOX can effectively describe the illumination change information, which meets the feature requirements of the illumination component extraction.

As the Figure 7 shows, our method is better than other methods in illumination correction of fabric images, and can effectively improve the visualization of detail features. BIMEFF, JED, EFF and LLNet can basically eliminate the influence of illumination, but there are still some regions where the brightness is too dark to extract the details effectively. The LIME makes the image over enhanced, which is not conducive to the extraction of detailed features, and reduces the contrast between the defect and the background.

### 4.2. Parameter Selection of the $L_{0}$ Gradient Minimization Method

We use the $L_{0}$ gradient minimization method to remove the background of dot-patterned fabric images. In the $L_{0}$ gradient minimization method, parameter λ affect the effect of defect detection. We explore the most appropriate parameter λ, and the results are shown in Figure 8.

As the Figure 8 shows, for dot-patterned fabric images, if the parameter λ is set too small, the background texture is hardly removed. Conversely, if the parameter λ is set too large, the defects are removed out.

In order to compare the impact of parameter λ on all kinds of dot-patterned fabric types, the parameter λ is 0.005, 0.01, 0.015, 0.02, 0.03, 0.04, and 0.05 respectively. Figure 9 shows the detection accuracy of four fabric types. It is proved by experiments that the selected number of λ is set between 0.005 and 0.05, which can meet the needs of four fabric types. As the Figure 9a shows, when λ=0.05, the broken end fabric will be mistakenly smoothed out. As the Figure 9d shows, when λ is set to 0.04 or 0.05, the thin bar fabric will be mistakenly smoothed out, and the accuracy rate cannot be calculated. When λ=0.02, the detection accuracy rate of the four types of defects is the best, so we set the parameter λ=0.02.

### 4.3. Generation of the Saliency Map

The saliency map of star-patterned fabric defect detection is shown in Figure 10. The saliency map of box-patterned fabric defect detection is shown in Figure 11. The saliency map of dot-patterned fabric defect detection is shown in Figure 12. As the Figure 10, Figure 11 and Figure 12 show, we can see that our method can effectively highlight the defect regions with saliency features, and it has strong adaptability and robustness to different types of defects.

### 4.4. Result Comparison

For each defect type of the fabric image database, an exemplar is randomly selected. The results of WGIS [11], Mobile-Unet [10], SHF [30], ER [14], CDPA [31], SR [15], and Ours are shown in Figure 13, Figure 14 and Figure 15.

For star-patterned exemplars shown in Figure 13, the detection accuracy of Ours on star-patterned fabrics are better than the rest in vision, the location and shape of defects are closest to the ground truth. WGIS and ER are basically unable to detect. For box-patterned exemplars shown in Figure 14, Mobile-Unet, SHF, CDPA, SR, and Ours can detect defects, but Ours is more closest to ground truth in the shape of defects. WGIS and ER cause a lot of false detection of defect-free points. For dot-patterned exemplars shown in Figure 14, all methods can detect defects, but the detection effect of Ours is more prominent.

### 4.5. Quantitative Comparison

In order to test the effectiveness of the method, we also made quantitative and qualitative comparisons. A number of metrics are used to evaluate the effectiveness of the method. That is, we calculted true positive (TP), false positive (FP), true negative (TN) and false negative (FN). According to the four parameters calculated above, true positive rate: TPR = TP/(TP + FN); false positive rate: FPR = FP/(FP+TN); positive predictive value: PPV = TP/(TP + FN); negative predictive value: NPV = TN/(TN + FN). Additionally, we also used the *f* value to evaluate the performance correctly.
(32)$$f=\frac{\left(\gamma^{2}+1\right)\times \mathrm{TPR}\times \mathrm{PPV}}{\mathrm{TPR}+\gamma^{2}\times \mathrm{PPV}},$$
where γ=1 in [37]. That is to say, Equation (Equation 32) can be rewritten as
(33)$$\begin{array}{cc} f & =\frac{2\times \mathrm{TPR}\times \mathrm{PPV}}{\mathrm{TPR}+\mathrm{PPV}}\\ & =2\times \left(\frac{1}{1/\mathrm{TPR}+1/\mathrm{PPV}}\right)\\ & =2\times \left(\frac{1}{(\mathrm{TP}+\mathrm{FN})/\mathrm{TP}+(\mathrm{TP}+\mathrm{FP})/\mathrm{TP}}\right)\\ & =2\times \left(\frac{\mathrm{TP}}{2\times \mathrm{TP}+\mathrm{FP}+\mathrm{FN}}\right) \end{array}.$$

The above Equations show that when the FN and FP increase, the value of *f* decreases, when FN and FP decrease, the value of *f* increases gradually and tends to 1. The *f* value only depands on TPR and PPV, which avoids the problem of incorrect evaluation and false inspection caused by the small defect regions. Consequently, we choose the *f* value as an important index to evaluate the performance correct of the method.

Table 1, Table 2 and Table 3 compare the quantitative results of the different algorithms (WGIS [11], Mobile-Unet [10], SHF [30], ER [14], CDPA [31], SR [15], and Ours) on star-, box- and dot-patterned fabrics. Besides, we marked the best results with black bold. The first to sixth columns are defect type name, TPR, FPR, PPV, NPV, and *f* value respectively. It should be noted that each row of the table is the average test result of a method, and the test results of all rows are classified according to the types of fabric defects.

For numeric star-patterned results in Table 1, our method get the highest overall TPR and PPV, while the overall *f* value is the highest, indicating that our method has achieved better overall recall and precision, and the detection accuracy rate is the best. For numeric box-patterned results in Table 2, our method get the highest overall TPR, NPV, and *f* value, indicating that the our method has the highest detection accuracy. Although Mobile-Unet achieves the lowest overall FPR and highest overall PPV, the overall TPR is only 60.75% and the *f* value is not the best, which is not conducive to actual detect. For numeric dot-patterned results in Table 3, our method get the highest overall TPR, NPV, and *f* value, indicating that the detection effect of our method is similar to box-patterned fabric. Although Mobile-Unet achieves the highest overall PPV, the overall TPR is only 64.88%. In summary, our method significantly improves the TPR and *f* value of star-, box- and dot-patterned fabric defect detection.

Figure 16 shows the TPR-PPV scatter plots of star-, box- and dot-patterned fabrics by seven methods, in which the same type of scatter plots represent different types of defects at different locations. In the scatter diagram, the closer the value of TPR and PPV is to 1 (100%), which indicates that the better the comprehensive detection effect of the method. The more centralized the scatter value distribution, the more robust and universal the performance of the method. As Figure 16 shows, in the scatter diagrams, our method is closest to the upper right corner of the diagrams, that is to say, the comprehensive TPR-PPV effect of our method is better. Besides, the scatter value of our method is the most aggregated, which shows that our method is more robust and adaptable to the detection of different patterns of fabrics.

### 4.6. Running Time Comparison

We compare our running time with Mobile-Unet [10], WGIS [11], ER [13], SR [15], SHF [30] and CDPA [31], as shown in Table 4.

As the Table 4 shows, compared with other methods based on image processing (WGIS, ER, SR, SHF and CDPA), our running time is significantly shorter. Mobile-Unet has the best real-time performance. However, Mobile-Unet needs to prepare a large number of defect images as training data in advance, and the training process also needs a lot of time.

## 5. Conclusions

This paper proposes a method of fabric defect detection based on illumination correction and visual salient features. In view of the limitations of traditional illumination correction methods, we propose a new method of illumination correction, which adjusts the brightness according to the illumination component in the global angle and enhances the contrast in the local angle. In order to eliminate the interference of background to the detection, the $L_{0}$ gradient minimization method is used to remove the texture background and highlight the defects. The traditional frequency domain based visual saliency detection algorithm only considers the saliency of the defects in the frequency domain. In this paper, the image is represented by quaternion image, and the two-dimensional fractional Fourier transform is used to enhance the saliency of the defects in the frequency domain and time domain. Finally, we use the region growing method and morphological processing to segment the saliency map and complete the defect detection. Experimental results on a standard database show that our method has better robustness and better detection effect than other methods. But it should be noted that our method has a high FPR for dot-patterned fabric defects, which needs to be improved in the future.

## Figures and Tables

**Figure 1 sensors-20-05147-f001:**
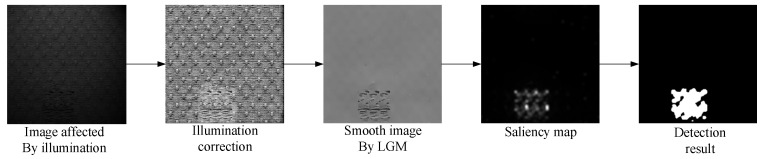
Framework of the proposed our method.

**Figure 2 sensors-20-05147-f002:**
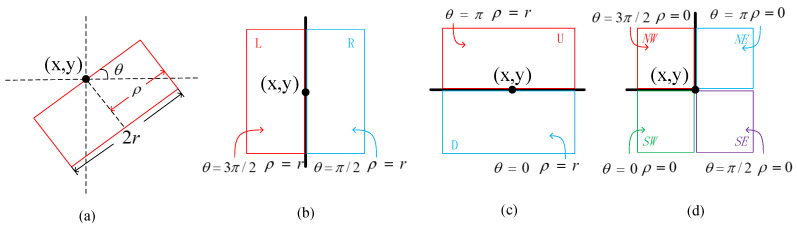
The definition of the side window. *r* is the radius of the window. (**a**) Definition of side window in continuous case. (**b**) The left (red rectangle) and right (blue rectangle) side windows. (**c**) The up (red rectangle) and down (blue rectangle) side windows. (**d**) The northwest (red rectangle), northeast (blue rectangle), southwest (green rectangle) and southeast (purple rectangle) side windows.

**Figure 3 sensors-20-05147-f003:**
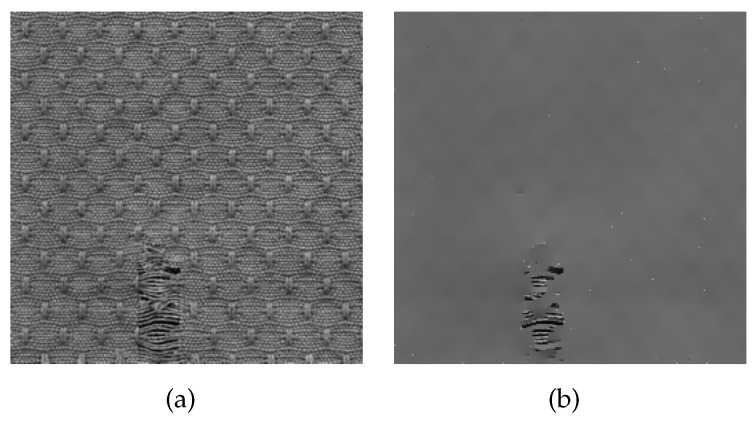
Smoothed background texture information via the $L_{0}$ Gradient Minimization (LGM). (**a**) Input fabric image; (**b**) Smoothed fabric image.

**Figure 4 sensors-20-05147-f004:**
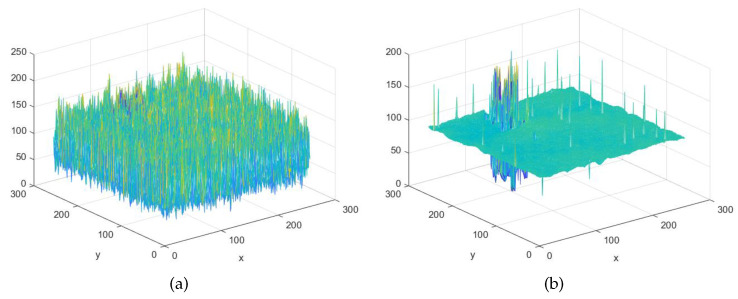
Implementation effect of the LGM. (**a**) Mesh diagram of fabric image; (**b**) Mesh diagram of smoothed fabric image.

**Figure 5 sensors-20-05147-f005:**
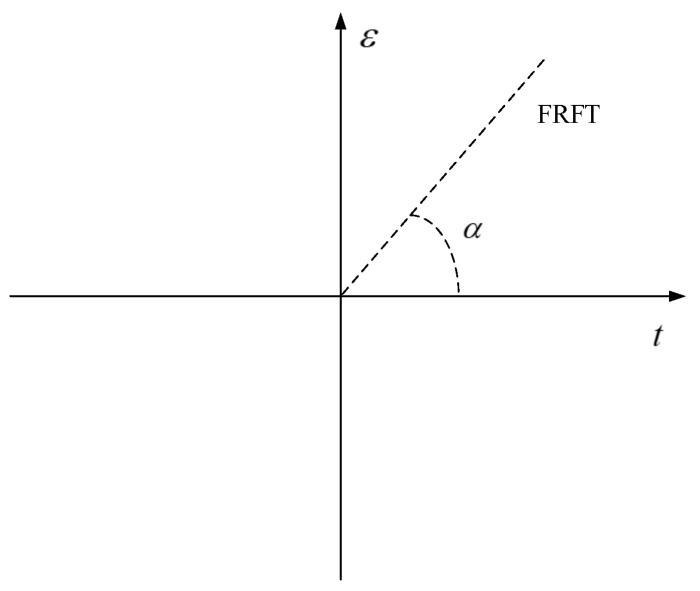
Transform domain of fractional Fourier transform.

**Figure 6 sensors-20-05147-f006:**
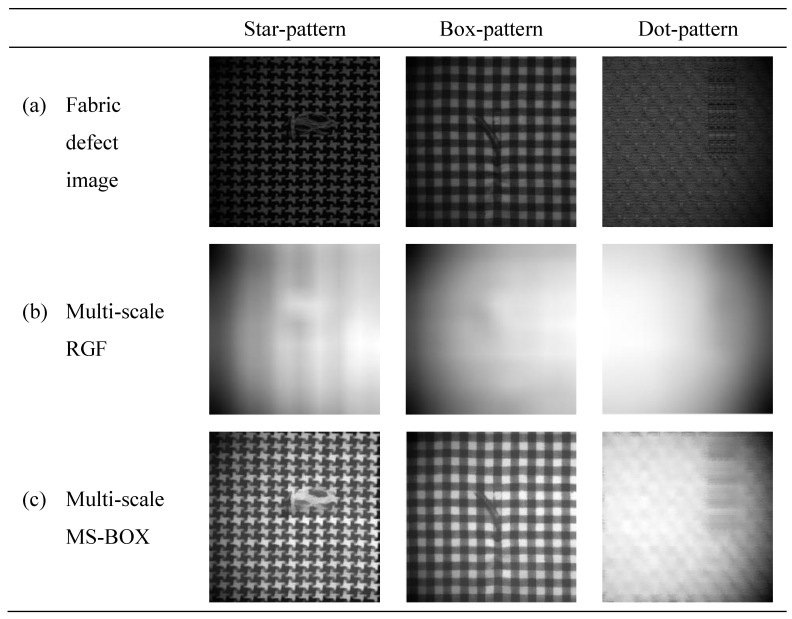
Comparison of illumination component extraction results.

**Figure 7 sensors-20-05147-f007:**
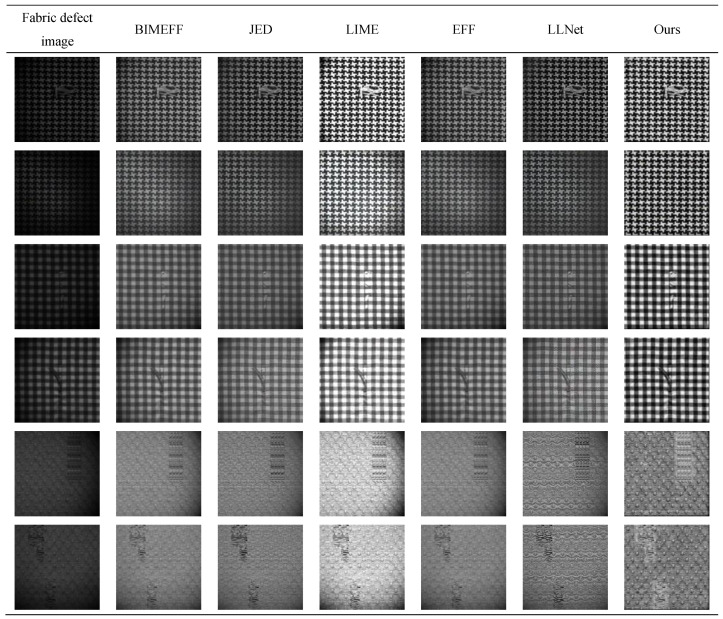
Comparison of different illumination correction methods.

**Figure 8 sensors-20-05147-f008:**
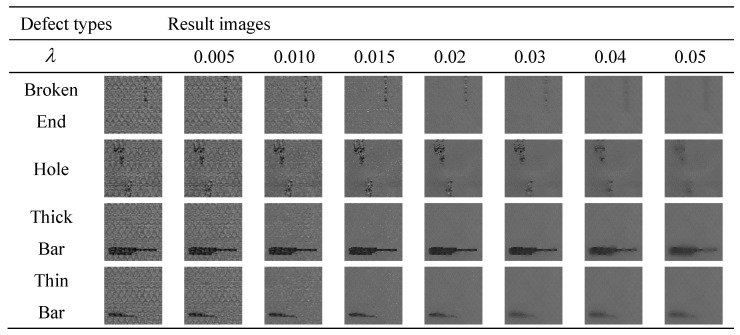
The optimum parameters of dot-patterned fabric types.

**Figure 9 sensors-20-05147-f009:**
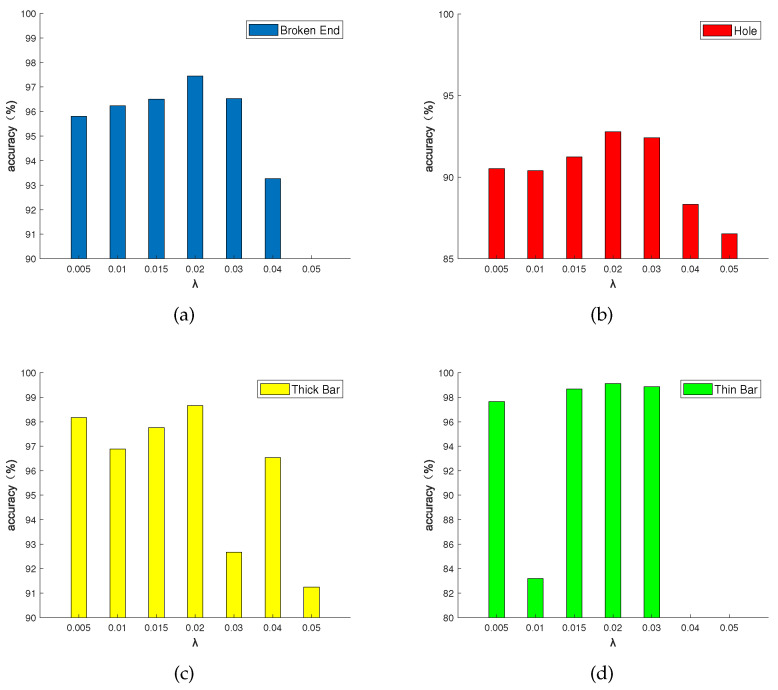
The influence of different parameter λ on the detection accuracy of four dot-patterned defect types. (**a**) broken end type; (**b**) hole type; (**c**) thick bar type; (**d**) thin bar type.

**Figure 10 sensors-20-05147-f010:**
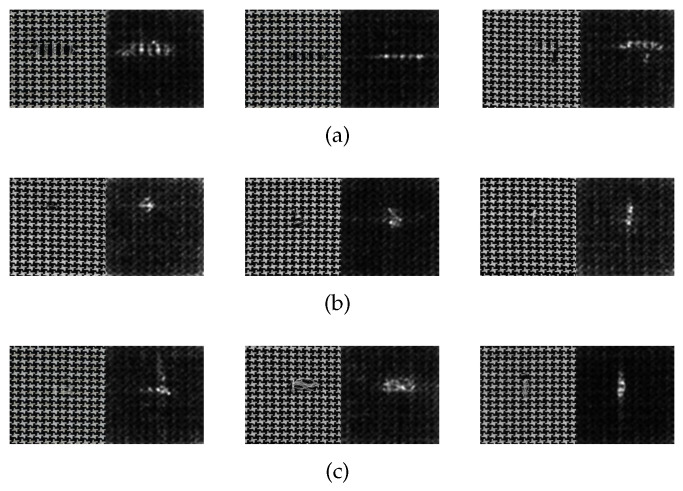
Saliency map of star-patterned fabric defect detection. (**a**) broken end type; (**b**) hole type; (**c**) netting multiple type.

**Figure 11 sensors-20-05147-f011:**
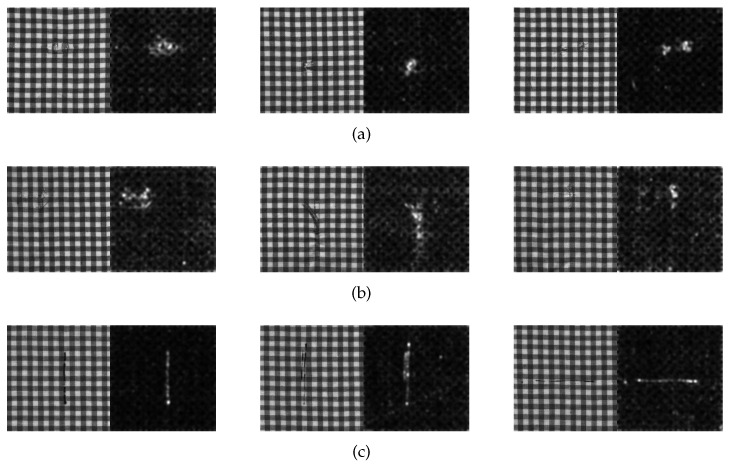
Saliency map of box-patterned fabric defect detection. (**a**) hole type; (**b**) netting multiple type; (**c**) thin bar type.

**Figure 12 sensors-20-05147-f012:**
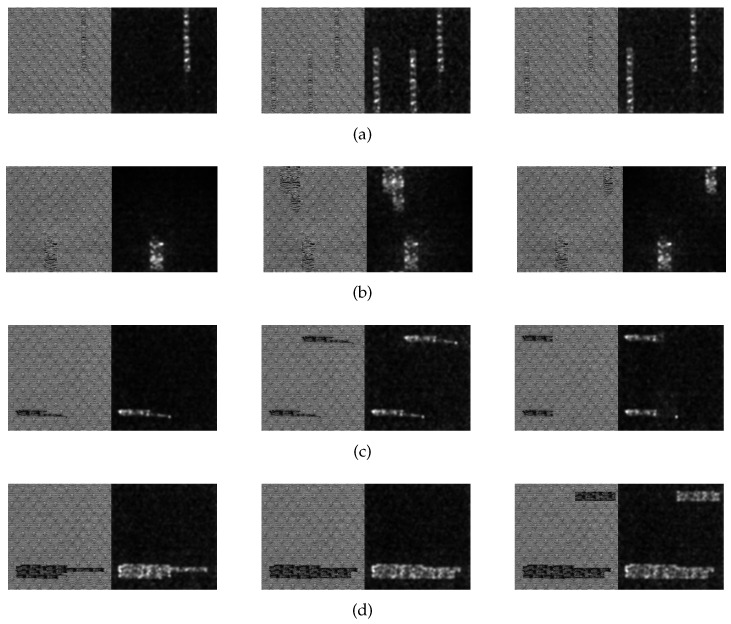
Saliency map of dot-patterned fabric defect detection. (**a**) broken end type; (**b**) hole type; (**c**) thin bar type; (**d**) thick bar type.

**Figure 13 sensors-20-05147-f013:**
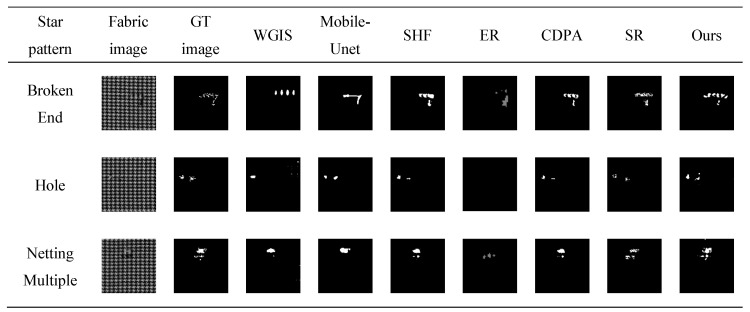
Each row depicts the detect inspection exemplars for 7 algorithms of a specific detection type. From top to bottom, these types are Broken End, Hole and Netting Multiple.

**Figure 14 sensors-20-05147-f014:**
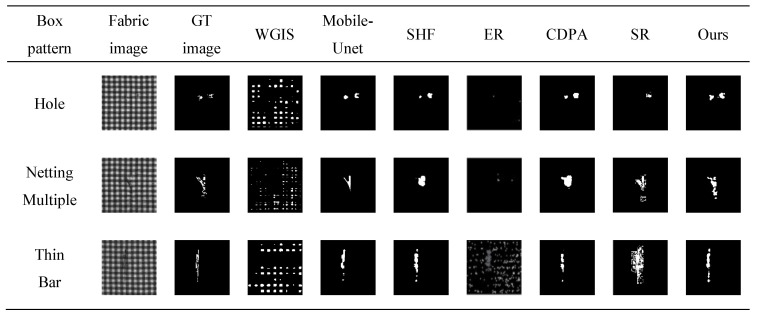
Each row depicts the detect inspection exemplars for 7 algorithms of a specific detection type. From top to bottom, these types are Hole, Netting Multiple and Thin Bar.

**Figure 15 sensors-20-05147-f015:**
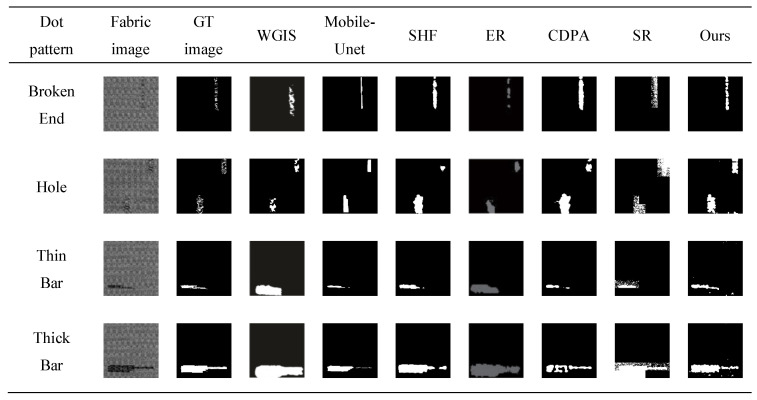
Each row depicts the detect inspection exemplars for 7 algorithms of a specific detection type. From top to bottom, these types are Broken End, Hole, Thin Bar and Thick Bar.

**Figure 16 sensors-20-05147-f016:**
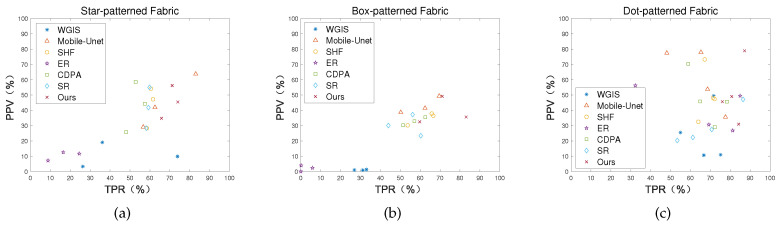
TPR-PPV scatter plots for (**a**) star-pattern, (**b**) box-pattern and (**c**) dot-pattern.

**Table 1 sensors-20-05147-t001:** Numerical results of each defect type for star-patterned fabric.

Star Pattern	TPR (%)	FPR (%)	PPV (%)	NPV (%)	*f* (%)	Methods
Broken End (5)	**73.88**	4.34	9.88	99.24	17.42	WGIS
	56.65	0.95	29.11	99.74	38.45	Mobile-Unet
	58.46	0.91	28.31	99.74	38.14	SHF
	8.79	1.16	7.17	99.27	7.89	ER
	48.05	**0.78**	25.82	99.62	33.59	CDPA
	58.16	2.67	28.40	99.64	38.16	SR
	65.81	0.79	**34.82**	**99.75**	**45.54**	Ours
Hole (5)	26.30	7.58	3.27	99.45	5.81	WGIS
	62.53	0.47	41.95	99.80	50.21	Mobile-Unet
	61.57	**0.46**	**47.22**	99.79	53.44	SHF
	24.47	1.23	11.68	99.54	15.81	ER
	57.51	0.47	44.28	99.78	50.03	CDPA
	59.26	4.60	41.80	99.78	49.02	SR
	**74.06**	0.51	45.48	**99.86**	**56.35**	Ours
Netting Multiple (5)	36.07	3.25	19.06	98.25	24.94	WGIS
	**83.01**	0.69	**63.77**	**99.77**	**72.12**	Mobile-Unet
	60.48	0.66	54.15	99.25	57.14	SHF
	16.42	0.82	12.61	98.54	14.26	ER
	52.84	0.62	58.63	99.16	55.58	CDPA
	59.80	0.79	55.03	99.43	57.31	SR
	71.21	**0.57**	56.22	99.17	62.83	Ours
Overall (15)	45.41	5.06	10.73	98.98	17.35	WGIS
	67.39	0.70	44.94	**99.77**	53.92	Mobile-Unet
	60.17	0.67	43.23	99.59	50.31	SHF
	16.56	1.07	10.48	99.12	12.83	ER
	52.80	**0.62**	42.91	99.52	47.34	CDPA
	59.07	2.68	41.74	99.61	48.91	SR
	**70.36**	0.63	**45.51**	99.59	**55.27**	Ours

**Table 2 sensors-20-05147-t002:** Numerical results of each defect type for box-patterned fabric.

Box Pattern	TPR (%)	FPR (%)	PPV (%)	NPV (%)	*f* (%)	Methods
Hole (5)	31.17	25.52	0.92	99.31	1.78	WGIS
	62.44	0.76	**41.41**	99.75	49.79	Mobile-Unet
	66.57	1.05	36.49	99.80	47.14	SHF
	0	**0.03**	0	97.69	0	ER
	62.60	0.97	35.55	99.72	45.34	CDPA
	56.20	0.80	37.20	99.67	44.76	SR
	**83.10**	1.33	35.67	**99.88**	**49.91**	Ours
Netting Multiple (5)	33.00	25.68	1.28	98.87	2.46	WGIS
	50.23	0.91	**38.65**	99.38	**43.68**	Mobile-Unet
	53.72	1.33	30.17	99.42	38.63	SHF
	0.15	**0.04**	4.00	95.81	0.28	ER
	51.38	1.52	30.28	**99.50**	38.10	CDPA
	44.00	0.16	30.10	99.36	35.74	SR
	**59.76**	1.44	32.50	99.46	42.10	Ours
Thin Bar (5)	26.90	24.20	1.02	99.07	1.96	WGIS
	69.57	**0.69**	**49.35**	99.70	57.74	Mobile-Unet
	65.81	1.05	37.86	99.67	48.06	SHF
	5.84	4.51	2.36	97.68	3.36	ER
	57.09	1.13	32.84	99.60	41.69	CDPA
	60.30	1.60	23.40	99.66	33.71	SR
	**71.10**	0.81	49.19	**99.72**	**58.14**	Ours
Overall (15)	30.35	25.13	1.07	99.08	2.06	WGIS
	60.75	**0.78**	**43.13**	99.61	50.44	Mobile-Unet
	62.03	1.14	34.84	99.63	44.61	SHF
	1.99	1.52	2.12	97.06	2.05	ER
	57.02	1.21	32.89	99.61	41.71	CDPA
	53.50	0.85	30.23	99.56	38.63	SR
	**71.32**	1.19	39.08	**99.68**	**50.49**	Ours

**Table 3 sensors-20-05147-t003:** Numerical results of each defect type for dot-patterned fabric.

Dot Pattern	TPR (%)	FPR (%)	PPV (%)	NPV (%)	*f* (%)	Methods
Broken End (5)	54.93	0.18	25.51	93.90	34.84	WGIS
	68.59	1.87	53.80	98.11	60.30	Mobile-Unet
	72.09	4.01	47.41	98.70	57.20	SHF
	32.27	**0.01**	**56.25**	91.90	41.01	ER
	78.32	5.20	45.65	98.94	57.68	CDPA
	53.36	26.50	20.30	82.60	29.41	SR
	**80.74**	5.09	49.05	**99.07**	**61.02**	Ours
Hole (5)	75.13	0.17	10.92	99.15	19.06	WGIS
	77.58	4.01	**35.63**	99.29	**48.83**	Mobile-Unet
	63.94	4.07	32.54	98.97	43.13	SHF
	69.21	**0.05**	30.63	98.94	42.46	ER
	72.18	4.82	29.04	99.04	41.41	CDPA
	61.17	6.50	22.28	98.95	32.66	SR
	**84.19**	5.38	30.95	**99.45**	45.26	Ours
Thick Bar (5)	71.66	0.17	49.46	96.19	58.52	WGIS
	65.26	0.27	77.97	95.01	71.05	Mobile-Unet
	67.13	2.30	73.27	93.11	70.15	SHF
	84.94	**0.15**	49.46	96.19	62.51	ER
	58.85	3.36	70.43	93.92	64.12	CDPA
	70.68	5.49	27.46	**99.23**	39.55	SR
	**87.18**	3.61	**78.95**	97.74	**82.86**	Ours
Thin Bar (5)	66.69	0.16	10.66	98.64	18.38	WGIS
	48.09	0.17	**77.47**	98.63	59.34	Mobile-Unet
	71.34	1.85	48.20	99.22	57.53	SHF
	81.22	**0.07**	26.81	99.30	40.31	ER
	64.88	1.87	45.78	99.02	53.68	CDPA
	**86.42**	16.58	47.15	97.43	**61.01**	SR
	76.02	2.36	45.67	**99.34**	57.06	Ours
Overall (20)	67.10	0.17	20.07	96.81	30.89	WGIS
	64.88	1.58	**61.22**	97.76	62.99	Mobile-Unet
	68.62	3.06	50.36	97.50	58.08	SHF
	66.91	**0.07**	40.79	96.58	50.68	ER
	68.56	3.81	47.72	97.58	56.27	CDPA
	67.90	13.76	29.30	94.55	40.93	SR
	**82.03**	4.11	51.16	**98.90**	**63.01**	Ours

**Table 4 sensors-20-05147-t004:** Comparison of running time by seven different methods.

Methods	Average Running Time/s	Hardware
Mobile-Unet [10]	0.021	One Nvidia TITAN Xp (GPU)
WGIS [11]	12.99	Intel Core i5-8300H (CPU)
ER [14]	12.13	Intel Core i5-8300H (CPU)
SR [15]	3.99	Intel Core i5-8300H (CPU)
SHF [30]	16.46	Intel Core i5-8300H (CPU)
CDPA [31]	10.43	Intel Core i5-8300H (CPU)
Ours	2.18	Intel Core i5-8300H (CPU)

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
