# Peer review of "Fabric Defect Detection Based on Illumination Correction and Visual Salient Features"

_sensors, 2020, doi:10.3390/s20185147_

Round 1

Reviewer 1 Report

The present manuscript can clearly be interesting to the readership of the journal. From a technical perspective, the manuscript is sound.

Here are the comments that should be taken into account as part of a revision:

  1. The very first sentence of the introduction reads: “The defect of fabrics will bring huge economic losses <...>”. Somehow it is difficult to accept that for what it is. Please reword this or provide explanation or motivation (e.g., an overview of the global fabrics market that should support your claim).
  2. All images presented in the manuscript show fabrics with clear periodic patterns. There are similar alternative methods that can be applied to defect detection based on pattern matching, such as, e.g., [https://doi.org/10.1016/j.patrec.2005.02.002]. Please compare your results to the ones obtained using such alternative approaches.
  3. One very important point is hardware versus software implementation of picture enhancement. The development of algorithms for your image processing pipeline is surely commendable and welcome, but one cannot help but wonder whether the initial part of the pipeline could have been solved on the picture taking side by simply considering better lighting conditions, i.e.: would it be possible to resolve the issues with illumination in the hardware sense by using specialized cameras or lighting solutions? In fact, please describe how the pictures of fabrics are taken in the first place.
  4. Please comment on the robustness of your solution. You claim that “<...> we can see that our method can effectively highlight the defect areas with saliency features, and it has strong adaptability and robustness to different types of defects.” However, which other undesired image features (apart from those stemming from lighting issues) can also result in false positives?

Overall, my suggestion is to work on the above items as part of a major revision: please also exhibit extensive proofreading efforts to improve the quality of presentation.

Author Response

Dear Reviewer

      Thank you very much for your valuable comments. We have studied the valuable comments from you carefully, and tried our best to revise the manuscript. The point to point responds to your comments is shown in the attachment. We have corrected and marked them in red in the revised manuscript.

Reviewer 2 Report

Authors presented image processing techniques to detect defects under illumination difficulties and reported performance in terms of accuracy and execution time. 

However, many deep learning based methods can solve this type of problem.

Authors need to explain the deep learning based methods for solving this type of problem and compare the performance between the proposed method and the deep learning based methods.

Author Response

(The authors gave the same response as above.)

Round 2

Reviewer 1 Report

I have no further comments.

Author Response

Dear Reviewer

        Thank you very much for your valuable comments. We are very grateful for review of our article.

Reviewer 2 Report

Although authors compared the proposed method with the oldest deep learning based method(U-net, '15), many deep learning based methods have been proposed after U-net.

I think authors need to include the comparison with at least DeepLabv3...

Furthermore, many deep learning based methods (such as ICnet) can be executed in real time on a GPU, while the proposed method required two seconds on a CPU.

I believe many researchers will use deep learning based methods to solve this type of problems, and thus authors need to explain the advantage of the image processing based method over the deep learning based methods.

Author Response

(The authors gave the same response as above.)
